# Model Predictive Direct Torque Control and Fuzzy Logic Energy Management for Multi Power Source Electric Vehicles

**DOI:** 10.3390/s22155669

**Published:** 2022-07-28

**Authors:** Khoudir Kakouche, Toufik Rekioua, Smail Mezani, Adel Oubelaid, Djamila Rekioua, Vojtech Blazek, Lukas Prokop, Stanislav Misak, Mohit Bajaj, Sherif S. M. Ghoneim

**Affiliations:** 1Laboratoire de Technologie Industrielle et de l’Information, Faculté de Technologie, Université de Bejaia, Bejaia 06000, Algeria; khoudir.kakouche@univ-bejaia.dz (K.K.); toufik.rekioua@univ-bejaia.dz (T.R.); adel.oubelaid@univ-bejaia.dz (A.O.); dja_rekioua@yahoo.fr (D.R.); 2Université de Lorraine, GREEN, F-54000 Nancy, France; smail.mezani@univ-lorraine.fr; 3ENET Centre, VSB—Technical University of Ostrava, 708 00 Ostrava, Czech Republic; lukas.prokop@vsb.cz (L.P.); stanislav.misak@vsb.cz (S.M.); 4Department of Electrical Engineering, National Institute of Technology, Delhi 110040, India; mohitbajaj@nitdelhi.ac.in; 5Department of Electrical Engineering, Graphic Era (Deemed to be University), Dehradun 248002, India; 6Department of Electrical Engineering, College of Engineering, Taif University, P.O. Box 11099, Taif 21944, Saudi Arabia; s.ghoneim@tu.edu.sa

**Keywords:** fuzzy logic, model predictive direct torque control, fuel cell, battery, permanent magnet synchronous motor, electric vehicle

## Abstract

This paper proposes a novel Fuzzy-MPDTC control applied to a fuel cell battery electric vehicle whose traction is ensured using a permanent magnet synchronous motor (PMSM). On the traction side, model predictive direct torque control (MPDTC) is used to control PMSM torque, and guarantee minimum torque and current ripples while ensuring satisfactory speed tracking. On the sources side, an energy management strategy (EMS) based on fuzzy logic is proposed, it aims to distribute power over energy sources rationally and satisfy the load power demand. To assess these techniques, a driving cycle under different operating modes, namely cruising, acceleration, idling and regenerative braking is proposed. Real-time simulation is developed using the RT LAB platform and the obtained results match those obtained in numerical simulation using MATLAB/Simulink. The results show a good performance of the whole system, where the proposed MPDTC minimized the torque and flux ripples with 54.54% and 77%, respectively, compared to the conventional DTC and reduced the THD of the PMSM current with 53.37%. Furthermore, the proposed EMS based on fuzzy logic shows good performance and keeps the battery SOC within safe limits under the proposed speed profile and international NYCC driving cycle. These aforementioned results confirm the robustness and effectiveness of the proposed control techniques.

## 1. Introduction

The use of electric vehicles (EVs) in the transportation sector has become a necessity in the last decade to deal with the energy crisis and environmental pollution problems, as they offer many advantages such as high efficiency, no carbon dioxide emissions, low maintenance, and no pollution [1]. EVs are mainly composed of fuel cells, supercapacitors, and batteries for the energy supply and storage part, as well as an electric motor for the traction part. As it is known, achieving the best performance EVs, requires an adapted EMS, to effectively regulate the flow of power between the different sources [2]. High-efficiency electric motors assisted by high-accuracy controllers are also still needed [3].

A proton exchange membrane fuel cell (PEMFC) is a promising source used to power vehicles due to its small size, low operating temperature, fast start-up, and high efficiency [4]. To overcome the drawback of slow dynamic response and to implement energy recovery, it is necessary to integrate energy storage sources, such as a supercapacitor and/or a Li-ion battery [5,6]. EMS is very important to manage energy allocation effectively, and its choice should be deeply investigated. In this context, several methods in the recent literature have been studied and evaluated, such as optimization methods, filter-based methods, controller methods, and rule-based methods. Optimization-based strategies that include Model Predictive Control [7], Grey Wolf Optimizer [8], Particle Swarm Optimization [9], etc, have been investigated in order to deal with complex management objectives (efficiency, cost, lifetime, etc). On the other hand, these strategies are complex and impose an important computation burden. Further filter-based EMSs can be found in the literature, including Low-Pass filters [10], and Wavelet Transform [11]. The filter-based management strategy aims to divide the required power into high- and low-frequency power as this strategy can improve the lifetime of the PEMFC stack. Nevertheless, the performance of the two EMSs depends strongly on the design of the filter which is a complicated task to be performed. Controller methods such as Backstepping [12], Passivity Control [13], Proportional-Integral Control [14], Sliding Control [15], etc, can obtain an exact calculation of the reference while taking into account the losses of the system. Rule-based strategies are mainly Fuzzy Logic, Artificial Neural Network [16], and State Machine [17]. Fuzzy logic-based EMS is widely used in fuel cell hybrid power systems [18,19,20,21,22]. This technique has the advantage to provide high performance and supporting imprecise system modeling. For these reasons, this technique is adopted in the current work.

Permanent magnet synchronous motors are widely considered the best type of electric motors that can be used to drive EVs. This is due primarily to their high efficiency, high power density, lightweight, and wide speed range [23]. With proper torque control, they can easily meet all of the vehicle requirements. Many torque control strategies, such as Field-Oriented Control (FOC) and Direct Torque Control (DTC), have been extensively researched in the literature [24,25]. DTC has a simple control structure and provides a fast dynamic torque response [26]. However, because it is based on hysteresis comparators, this technique has some drawbacks such as important torque, flux ripples, and variable switching frequency. Many methods have been proposed to mitigate these drawbacks. In [27], the authors proposed a strategy named Space Vector Modulation DTC (SVM-DTC) based on a constant switching frequency, and the results showed good performance. However, this strategy requires a precise design of the PI controller and system parameters. In [28], a multilevel inverter was used, which increases the number of voltage vectors. The simulation results indicated good performances by reducing torque and flux ripples. Nevertheless, issues with complexity and switching losses appeared. Fuzzy logic and artificial neural networks are also used as artificial intelligence controllers [29,30]. Authors in [31,32,33,34] have proposed a Model Predictive Direct Torque Control strategy that is based on predicting the control variables such as flux and torque while minimizing the error of the predicted control variables; this improves control accuracy while maintaining the control system’s simplicity.

To cope with the aforementioned challenges and improve EV, two control strategies are proposed in this work. The first one is based on fuzzy logic, applied to the Li-ion battery-PEMFC hybrid power system. The second one is based on torque predictive control applied to the PMSM. Results obtained using Matlab/Simulink and those obtained using a RT LAB simulator have clearly shown the effectiveness of the proposed control techniques under different driving modes (cruising, acceleration, idling, and regenerative braking). In order to properly situate this work, the main contributions made through this paper are:An adequate EMS strategy based on fuzzy logic control is developed to ensure vehicle propulsion power and to regulate efficiently the energy flow of the power sources.A model predictive direct torque control strategy is proposed to control the vehicle traction machine with the objective of minimizing torque and flux ripples and ensuring satisfactory speed tracking.A detailed physical model of the EV (the vehicle dynamics system, the electric power system, and control system) is established under Matlab/Simulink environment.Real-time simulation using the RT LAB platform is performed to confirm the obtained simulation results.

The results obtained using Matlab/Simulink as well as the experimental ones obtained using the RT LAB simulator are presented, and the main conclusion of this work summarizes and proves proposed strategies.

## 2. Electric Vehicle Description and Modeling

The EV general configuration is shown in Figure 1. It consists of a hydrogen tank with a flow regulator, a PEMFC stack as a primary power source, a Li-ion battery as a secondary power source, a PMSM, and a control system. The entire VE system’s construction can be divided into two parts:

On the sources side, the flow rate regulator adjusts the power of the PEMFC stack and regulates the pressure of the hydrogen flow. The PEMFC stack is connected to the DC bus voltage via a unidirectional in-current DC-DC boost converter to DC bus voltage. The excessive electricity generated by the PEMFC stack is used to charge the battery. The Li-ion battery is connected to the DC bus voltage via a bidirectional DC-DC buck-boost converter to recover the braking energy and supply power.

On the traction side, PMSM of 50 kW rated power, fed by a two-level inverter, converts the electric power coming from the two power sources into mechanical power.

The control system integrates the control of both traction machine and power sources.

### 2.1. Fuel-Cell Model

The voltage of the PEMFC stack $V_{FC}$ is given by [18,35]:(1)$$V_{FC}=E-RI_{FC}$$
(2)$$E=E_{OC}-NA\ln\left(\frac{I_{FC}}{i_{0}}\right)\cdot \frac{1}{\frac{sT_{d}}{3}+1}$$
where $V_{FC}$ and $I_{FC}$ are the voltage and current of the PEMFC stack, respectively, R is the internal resistance, $E_{OC}$ is the open circuit voltage, $i_{0}$ is the exchange current, N, A and $T_{d}$ are the cells number, the tafel slope, and the response time, respectively.$\;E_{OC}$, $i_{0}$ and A are given by:(3)$$E_{OC}=K_{C}E_{N}$$
(4)$$i_{0}=\frac{zFk\left(P_{H_{2}}+P_{O_{2}}\right)}{Rh}e^{\frac{-\Delta G}{RT}}$$
(5)$$A=\frac{RT}{z\alpha F}$$
where $K_{C}$ and $E_{N}$ are the voltage constant at nominal condition of operation and Nernst voltage respectively, z is the number of moving electrons (z=2), F, R, k, h and T are the Faraday’s constant, the ideal gas constant, the Boltzmann’s constant, the Planck’s constant and the temperature of operation respectively, ΔG is the activation energy barrier, and α is the charge transfer coefficient.

The PEMFC stack includes hydrogen controller and oxygen controller, which regulate the flows rate of $H_{2}$ and $O_{2}$, respectively. The utilization rates of $H_{2}$ and $O_{2}$ are calculated as:(6)$$Uf_{H_{2}}=\frac{60000RTNI_{FC}}{zFP_{fuel}V_{fuel}x\%}$$
(7)$$Uf_{o_{2}}=\frac{60000RTNI_{FC}}{2zFP_{air}V_{air}y\%}$$
where $P_{fuel}$ and $P_{air}$ are the absolute supply pressure of fuel and absolute air, respectively, $V_{fuel}$ and $V_{air}$ are the fuel flow rate and the air flow rate, respectively, x% and y% are the percentage of $H_{2}$ in the fuel and $O_{2}$ in the oxidant, respectively.

The parameters of the used PEMFC stack are given in Table 1.

### 2.2. Battery Model

Li-ion batteries are used in this work due to their high energy density, high efficiency, and long lifetime when compared to other battery types such as (NiCd, lead-acid, or NiMH) [36].

The Li-ion battery voltage can be calculated using two different equations [36,37].
(8)$$V_{disch\arg e}=E_{0}-R\cdot i-K\frac{Q}{Q-it}\cdot (it+i*)+A\exp\left(-B\cdot it\right)$$
(9)$$V_{ch\arg e}=E_{0}-R\cdot i-K\frac{Q}{it-0.1Q}\cdot i*-K\frac{Q}{Q-it}\cdot it+A\exp\left(-B\cdot it\right)$$
where R, K, Q, $E_{0}$, it, $i^{*}$, A and B are the Li-ion battery internal resistance, the polarization constant, the Li-ion battery capacity, the Li-ion battery constant voltage, the actual Li-ion battery charge, the filtered Li-ion battery current, the exponential zone amplitude, and the exponential zone time constant inverse, respectively.

The Li-ion battery state of charge can be determined using Equation (10).
(10)$$SOC_{bat}=100\left(1-\int \frac{i_{\left(t\right)}dt}{Q}\right)\;$$

### 2.3. Permanent Magnet Synchronous Motor Model

The mathematical model of PMSM in the *d**-q* rotor reference frame can be expressed as follows in Equations (11) and (12) [23]:(11)$$V_{sd}=R_{s}I_{sd}+\frac{d\phi_{sd}}{dt}+\omega \phi_{sq}$$
(12)$$V_{sq}=R_{s}I_{sq}+\frac{d\phi_{sq}}{dt}+\omega \phi_{sd}$$
where the totalized flux $\phi_{sd}$
*and*
$\phi_{sq}$ are given by: (13)$$\phi_{sd}=L_{sd}I_{sd}+\phi_{f}$$
(14)$$\phi_{sq}=L_{sq}I_{sq}$$

The electromagnetic torque expression is given by Equation (15):(15)$$T_{e}=\frac{3}{2}pI_{sq}\left(\left(L_{sd}-L_{sd}\right)I_{sd}+\phi_{f}\right)$$

The PMSM mechanical equation is given by:(16)$$J\frac{d\Omega }{dt}=T_{e}-T_{r}-f\Omega$$

The estimations of the torque, flux, and the load angle can be expressed by the following set of equations:(17)$$\overset{\wedge }{T_{e}}=\frac{3}{2}p\left(I_{sq}\phi_{sd}-I_{sd}\phi_{sq}\right)$$
(18)$$\overset{\wedge }{\phi_{s}}=\sqrt{{\left(\phi_{sd}\right)}^{2}+{\left(\phi_{sq}\right)}^{2}}$$
(19)$$\overset{\wedge }{\theta }=\tan^{-1}\frac{\phi_{sq}}{\phi_{sd}}$$

Motor parameters [38] are summarized in Table 2.

### 2.4. Vehicle Dynamics System

The dynamic model of the electrical vehicle is depicted in Figure 2. The entire mechanical part (longitudinal vehicle dynamics, viscous friction, differential, tires, and reduction gear) of the vehicle dynamic system is modeled by Souleman Njoya Motapon and Louis-A. Dessaint [35]. The longitudinal vehicle dynamics block takes into account body mass, aerodynamic drag, and weight distribution between axles. Meanwhile, wind speed and road inclination are not considered in this model.

The parameters of the used vehicle [38] are given in Table 3.

## 3. System Control

### 3.1. Energy Management Strategy

To satisfy the load power demand and to ensure an efficient power distribution of the electric vehicle power system, an appropriate EMS is required [2]. These objectives can only be met by controlling the power response of each energy source according to the load demand. We used a fuzzy logic control-based EMS because it is flexible, efficient, and works well without exact mathematical models.

#### 3.1.1. Input and Output Parameters

The inputs parameters of the fuzzy logic controller are the Li-ion battery state-of-charge SOC and the load power (Pload) obtained by multiplying the motor speed and the required motor torque, and the output parameter is the reference power of the fuel cell, as illustrated in Figure 3.

The fuzzy set for Li-ion battery SOC is divided into “Low” (L), “Medium” (M), and “High” (H). The fuzzy set for Pload is also divided into “Negative” (N), “Very Low” (VL), L, M, H, “Very High” (VH), indicating the power demand from low to high levels. The fuzzy set for Pfc is classified as “Zero” (ZE), VL, L, M, H, VH. The gap between (ZE) and (VL) of Pfc in Figure 4c is the fuel cell system’s low-efficiency zone (including the cooling fan, humidifier, and other accessories), therefore the PEMFC stack should not operate in this range. The triangular and trapezoidal membership functions (MFs) are used in this case as shown in Figure 4. The choice of the triangular and trapezoidal shapes is to reduce the variations in the generated power reference.

#### 3.1.2. Fuzzy Inference Rules

The fuzzy logic rules corresponding to this energy management are designed and presented in Table 4. The choice of the rules is according to the desired operation on the PEMFC stack. For example, when the SOC of the Li-ion battery is low and the power demanded by the vehicle is very high, then the power that the PEMFC stack must provide will be very high. In the same way, the PEMFC stack provides a power that is lower than the power demand when the Li-ion battery is highly charged because in this case, the Li-ion battery must be solicited to reduce it to a medium charge in order to take advantage of the energy recovery from braking. Based on this general idea, the choice of these rules remains arbitrary according to the desired functioning, while respecting the response time of the two sources. The SOC of the Li-ion battery should be maintained between 30% and 80%. This procedure prevents deep discharging and overcharging, which can reduce the lifespan of a Li-ion battery. The fuzzy rules surface is shown in Figure 5.

The proposed fuzzy logic controller uses the Mamdani inference procedure, with the centroid method for defuzzification [36].

### 3.2. DC Bus Voltage Regulation and PEMFC Stack Converter Control

The electrical energy sources used to supply the EV must be well controlled via the converters by regulating their currents and/or output voltages. The converters connected to the energy sources control the output power and voltage [39]. The regulation of the DC bus voltage and the control of the PEMFC stack power are presented in Figure 6. PEMFC stack and Li-ion battery are, respectively, connected to the DC bus via unidirectional and bidirectional DC-DC converters. The power management block generates the reference current $I_{FC}^{*}$ which has to be limited in a slope in order to respect the constraints related to the PEMFC stack dynamics. A PI controller is used to control the PEMFC stack power by adjusting the current to its reference value $I_{FC}^{*}$. The Li-ion battery regulates the DC bus voltage by tracking the reference voltage $V_{DC\_ref}$. A double PI regulation loop is used to maintain the DC bus voltage close to its reference and to control the Li-ion battery power.

### 3.3. Model Predictive Direct Torque Control

The model predictive direct torque control strategy basic principle is to predict the system future behavior over time using the PMSM model [33]. This strategy is used to control both the flux and torque of the PMSM in the EV system. The numerical implementation of the MPDTC algorithm for PMSM in the EV can be divided into two steps. Step 1 is to predict the controlled variables, and the second step is to select the voltage vector to be applied in the next sampling time. To determine the best voltage vector to use in the next sampling time, a cost function is constructed. The optimal voltage vector is chosen based on the objectives with the minimum error, resulting in reduced ripples. In MPDTC, the controlled variables are predicted using Forward Euler approximation [40].

The block diagram and the flowchart of MPDTC are illustrated in Figure 7 and Figure 8.

#### 3.3.1. Current Prediction

Stator current prediction can be expressed in discrete time steps using Equations (20) and (21) [41]:(20)$$I_{sd}\left(k+1\right)=\left(1-\frac{R_{s}T_{s}}{L_{sd}}\right)I_{sd}\left(k\right)+\frac{\omega \left(k\right)L_{sq}T_{s}}{L_{sd}}I_{sq}\left(k\right)+\frac{T_{s}}{L_{sd}}V_{sd}\left(k\right)$$
(21)$$I_{sq}\left(k+1\right)=\left(1-\frac{R_{s}T_{s}}{L_{sq}}\right)I_{sq}\left(k\right)-\frac{\omega \left(k\right)L_{sd}T_{s}}{L_{sq}}I_{sd}\left(k\right)-\frac{\omega \left(k\right)\phi_{f}T_{s}}{L_{sq}}+\frac{T_{s}}{L_{sq}}V_{sq}\left(k\right)$$
where $T_{s}$ is the sampling time.

#### 3.3.2. Flux and Torque Predictions

Stator flux prediction in *d**-**q* frame is expressed using Equations (22)–(24) shown below:(22)$$\phi_{sd}\left(k+1\right)=L_{sd}I_{sd}\left(k+1\right)+\phi_{f}$$
(23)$$\phi_{sq}\left(k+1\right)=L_{sq}I_{sq}\left(k+1\right)$$
(24)$$\phi_{s}\left(k+1\right)=\sqrt{{\left(\phi_{sd}\left(k+1\right)\right)}^{2}+{\left(\phi_{sq}\left(k+1\right)\right)}^{2}}$$

Based on the predicted current and flux, the electromagnetic torque is calculated using Equation (25):(25)$$T_{e}\left(k+1\right)=\frac{3}{2}p\left(\phi_{sd}\left(k+1\right)I_{sq}\left(k+1\right)-\phi_{sq}\left(k+1\right)I_{sd}\left(k+1\right)\right)$$

#### 3.3.3. Cost Function Minimization

The cost function includes absolute torque and flux error values; this function is chosen so that the voltage vector chosen produces the best flux and torque control. The cost function is defined as follows:(26)$$g=\left|T_{e}^{*}-T_{e}\left(k+1\right)\right|+\gamma \left|\phi_{s}^{*}-\phi_{s}\left(k+1\right)\right|$$
where $T_{e}^{*}$ is the reference values of torque, $\phi_{s}^{*}$ is the reference values of stator flux, $T_{e}\left(k+1\right)$ is the predicted values of torque, $\phi_{s}\left(k+1\right)$ the predicted values of stator flux, and γ is the weighting factor.

#### 3.3.4. Time Delay Compensation

The voltage vector selected at time (*k*) will be applied at time (*k* + 1) with a delay of one-step. Because the sampling time is very small, this deteriorates the system performance. Therefore, a compensation of the delay time must be performed to improve the system performance. Therefore, the prediction by two steps ahead (*k* + 2) will be considered. The prediction of the stator currents at time (*k* + 2) can be expressed using Equations (27) and (28) [31,42]:(27)$$I_{sd}\left(k+2\right)=\left(1-\frac{R_{s}T_{s}}{L_{sd}}\right)I_{sd}\left(k+1\right)+\frac{\omega \left(k+1\right)L_{sq}T_{s}}{L_{sd}}I_{sq}\left(k+1\right)+\frac{T_{s}}{L_{sd}}V_{sd}\left(k+1\right)$$
(28)$$I_{sq}\left(k+2\right)=\left(1-\frac{R_{s}T_{s}}{L_{sq}}\right)I_{sq}\left(k+1\right)-\frac{\omega \left(k+1\right)L_{sd}T_{s}}{L_{sq}}I_{sd}\left(k+1\right)-\frac{\omega \left(k+1\right)\phi_{f}T_{s}}{L_{sq}}+\frac{T_{s}}{L_{sq}}V_{sq}\left(k+1\right)$$

The cost function g can be extended by adding the current magnitude limitation to avoid overcurrent. This term is defined as follows in Equation (29) [32]:(29)$$\overset{\wedge }{f}\left(I_{sd}\left(k+2\right),I_{sq}\left(k+2\right)\right)=\left\{ \begin{array}{l} \begin{array}{cc} \infty & \begin{array}{c} \begin{array}{cc} if & \left|I_{ds}\left(k+2\right)\right| \end{array}>I_{\max}\\ \begin{array}{cc} or & \left|I_{qs}\left(k+2\right)\right|>I_{\max} \end{array} \end{array} \end{array}\\ \begin{array}{cc} 0 & \begin{array}{c} \begin{array}{cc} if & \left|I_{ds}\left(k+2\right)\right|\leq I_{\max} \end{array}\\ \begin{array}{cc} or & \left|I_{qs}\left(k+2\right)\right|\leq I_{\max} \end{array} \end{array} \end{array} \end{array}\right.$$

The total cost function *g* of MPDTC with compensation of the computation time delay is given as:(30)$$\begin{array}{c} g=\left|T_{e}^{*}-\frac{3}{2}p\left(\phi_{sd}\left(k+2\right)I_{sq}\left(k+2\right)-\phi_{sq}\left(k+2\right)I_{sd}\left(k+2\right)\right)\right|+\gamma \left|\phi_{s}^{*}-\sqrt{{\left(\phi_{sd}\left(k+2\right)\right)}^{2}+{\left(\phi_{sq}\left(k+2\right)\right)}^{2}}\right|\\ +\\ \overset{\wedge }{f}\left(I_{sd}\left(k+2\right),I_{sq}\left(k+2\right)\right) \end{array}$$

## 4. Simulation Results and Discussion

In order to evaluate EV efficacy and to test the dynamic performance of the proposed strategies, numerical simulations in MATLAB/Simulink environment have been performed. A driving cycle under different operating modes is proposed as shown in Figure 9.

During the phase [0 s–4 s], (starting), the pedal position is set to 70% and the EV increases the speed to 25 km/h.During the phase [4 s–8 s], (cruising), the pedal position is released to 10% and the EV increases its speed to 26 km/h.During the phase [8 s–13 s], (accelerating), the pedal position is depressed to 85% and the EV increases the speed from 26 to 60 km/h.During the phase [13 s–14 s], (idling), the pedal position is set to 60% and the EV speed decreases to 58.5 km/h.During the phase [14 s–16 s], (regenerative braking), the pedal position is set to −70% and the speed of EV drops to 43 km/h.

Figure 10 shows the simulation results of the traction chain under different driving modes using the MPDTC strategy where (a) is the stator current of the PMSM, (b) Vehicle speed response, (c) Electromagnetic torque of the PMSM, and (d) is the stator flux. In Figure 10a, it is noticed that the variation of the stator current dynamics corresponds to the changes in the acceleration. The zoom shows that the stator current is perfectly sinusoidal. In Figure 10b, it is clearly seen that the vehicle speed depends on the variation of the pedal acceleration. Figure 10c shows the behavior of the electromagnetic torque, where it can be seen that the measured torque perfectly follows the reference torque even when the trajectories change abruptly with a good ripple, demonstrating the effectiveness of the MPDTC technique used on the system. In Figure 10d, it can be seen that the measured stator flux follows its reference with good precision during all trajectories with a good ripple.

To check the MPDTC performances in a comparative way, and to shed light on the obtained results, a comparison between the proposed MPDTC and the conventional DTC was made and is presented in Table 5, Figure 11a,b and Figure 12a,b. The electromagnetic torque zoom is presented in Figure 11. A reduction of 54.54% in torque ripples was observed compared to conventional DTC. Figure 12 depicts the stator flux zoom. An improvement of 77% compared to conventional DTC can be seen.

The fast Fourier transform (FFT) analyses of phase current and THD calculation for the proposed MPDTC and conventional DTC are presented in Table 5 and Figure 13a,b. The THD of MPDTC is 1.45%, which is lower than that of conventional DTC which is 3.13%, which presents an improvement of 53.37%. The simulation results presented in Table 5 confirm the performance of the proposed MPDTC is superior to that of conventional DTC.

Figure 14a depicts the Li-ion battery SOC. It can be clearly observed that the Li-ion battery SOC decreases slightly from 60% to 58.7% between [0 s, 13 s], then the Li-ion battery SOC increases until it reaches 59.4%, after absorbing the braking Energy. The DC bus voltage of the electric vehicle is depicted in Figure 14b. It can be seen that the measured voltage is well regulated to its reference (500 V) regardless of load demand. The zoom shows an overshoot in DC bus voltage of (5 V), this allows us to say that the DC bus voltage control is satisfactory.

Figure 15 shows the power distribution of the electric vehicle under different driving modes with an initial Li-ion battery SOC of 60% for the EMS based on fuzzy logic. When the vehicle is accelerating, the Li-ion battery alone powers the vehicle until t = 2 s. From 2 s to 4 s, the power demand increases slightly, the PEMFC stack begins to operate, providing a significant portion of the driving power, and the Li-ion acts as a secondary power source. From 4 s to 8 s the power demand decreases, the PEMFC stack provides no power, and the Li-ion battery alone powers the vehicle. From 8 s to 13 s the power demand is higher, the PEMFC stack is the primary power source, and the Li-ion battery becomes a secondary source, with the Li-ion battery SOC generally declining. From 13 s to 14 s no power is demanded, and the Li-ion battery and the PEMFC stack provide no power. From t = 14 s the regenerative braking mode starts, the Li-ion battery absorbs the kinetic braking energy and the motor acts as a generator.

Figure 16 shows the performance of the electric vehicle under different driving modes and battery states of charge. In Figure 16a, the initial Li-ion battery SOC is high (SOC = 80%); in this situation, the Li-ion battery provides a significant portion of the driving power, and the PEMFC stack only operates when the power demand becomes high, to protect the Li-ion battery from overcharging and keep the SOC in a relatively low level. In Figure 16b, the initial Li-ion battery SOC is low (30%); in this situation, the PEMFC stack is configured to generate more power than the engine power. To charge the Li-ion battery to acceptable levels, and to protect the Li-ion battery from deep discharges, the Li-ion battery only discharges in high acceleration to supplement the power demand.

Figure 15 and Figure 16 clearly show that the EMS based on fuzzy logic ensures a convenient power flow and gives good performance with the changes in driving conditions, and battery states of charge.

### Driving Cycle Test

The New York City Cycle (NYCC) driving cycle is adopted to analyze the performance of the proposed EMS based on fuzzy logic. Three different scenarios are defined to test the performance of EV under different states of charge of the Li-ion battery. The three scenarios are as follows:

Scenario 1: the initial SOC of the Li-ion battery is 60% when starting.

Scenario 2: the initial SOC of the Li-ion battery at startup is 80%.

Scenario 3: the initial SOC of the Li-ion battery is 30%.

Figure 17 shows the power curves for the PEMFC stack, Li-ion battery, motor, and Li-ion battery SOC in scenario 1. It can be seen in Figure 17a, that the PEMFC stack does not operate in the low power range [0–5 kW] and becomes active in the high power demand where the efficiency of the PEMFC system is relatively high. The Li-ion battery is configured to provide all the charging power in the low power demand range, assist the fuel cell in the high power demand range, and absorb the braking energy when the vehicle decelerates. The final SOC value of the Li-ion battery in this scenario is 59.5%, as seen in Figure 17a.

Figure 18 shows the power curves and the SOC of the Li-ion battery in the second scenario. As seen in Figure 18a, the Li-ion battery is regulated to provide a large amount of power to protect it from overcharging, and the PEMFC stack intervenes in the high power demand. The Li-ion battery SOC decreases from 80% to 74% as shown in Figure 18b.

Figure 19a shows the power curves in the third scenario. The fuel cell is operated to provide power for the load and charge the Li-ion battery to protect it from deep discharges. The Li-ion battery SOC increases from 30% to 46.5%, as shown in Figure 19b.

The results show that the proposed energy management strategy based on fuzzy logic is robust and could improve the efficiency of the PEMFC stack, mainly because this proposed EMS makes the PEMFC stack operate in the high-efficiency region.

## 5. Real-Time Platform Using RT-LAB

The real-time simulation of the proposed control is implemented in this section using the discrete real-time simulator, the RT LAB platform. Figure 20a depicts the real-time simulation bench set up in the LTII laboratory of Bejaia, which consists of the following components: (1) a host PC, (2) an OP5700 real-time digital simulator, (3) a HIL controller and data acquisition interface OP8660, and (4) a digital oscilloscope. As it is shown in Figure 20b, the first step toward real-time simulation is the model separation. The EV system is split into computation and console blocks. Blocks that contain computations such as energy management strategy, MPDTC, vehicle dynamics, motor and power sources models are placed on the computation subsystem, which is constituted of a master (SM) bloc highlighted with red dashed lines and a slave (SS) block. Scopes and constants are placed in the console block. In RT-LAB, each computation subsystem is assigned to a different core. In other words, each subsystem is coded in C and built for execution among its processors using Mathworks code generator Real-Time-Workshop (RTW) [43]. After compilation, the code is loaded into target OP5700 via TCP/IP protocol and executed via parallel processing. Finally, all observations in display blocks are monitored and displayed on the digital oscilloscope via I/O channels.

A real-time simulation was performed after the EV simulation system was decomposed and adapted for use in the RT LAB platform.

In order to evaluate the two techniques proposed in this work, namely the MPDTC on the motor side, and the EMS based on fuzzy logic on the source side, a driving cycle under different operating modes has been applied (Figure 9). Figure 21, Figure 22, Figure 23 and Figure 24 present experimental results of the two techniques proposed. The results obtained using a discrete real-time simulator RT LAB are very close to the simulation results with the same remarks previously mentioned in the simulation results. Experimental results prove the effectiveness of the proposed control techniques.

## 6. Conclusions

In this paper, the authors focused on improving the performance of an EV by introducing a Fuzzy-MPDTC-based control. This control is divided into two parts: The first part is dedicated to the control of the traction machine by introducing a MPDTC technique for PMSM control. The second part proposes a fuzzy logic-based EMS for the electric vehicle power system. To evaluate these techniques, a driving cycle under different operating modes has been proposed. The main conclusions are listed as follows. On the motor side, several objectives are achieved by applying a predefined cost function, the electromagnetic torque response follows its reference with small ripple values, a 54.54% improvement compared to the classical DTC; the current ripple is reduced and the reference tracking is ensured. On the sources side, the fuzzy logic-based EMS provides robust performance under various battery states of charge and rapid variation in power demand. Real-time simulation results were obtained using a discrete real-time simulator; the RT LAB platform confirmed the effectiveness and robustness of the proposed control techniques.

## Figures and Tables

**Figure 1 sensors-22-05669-f001:**
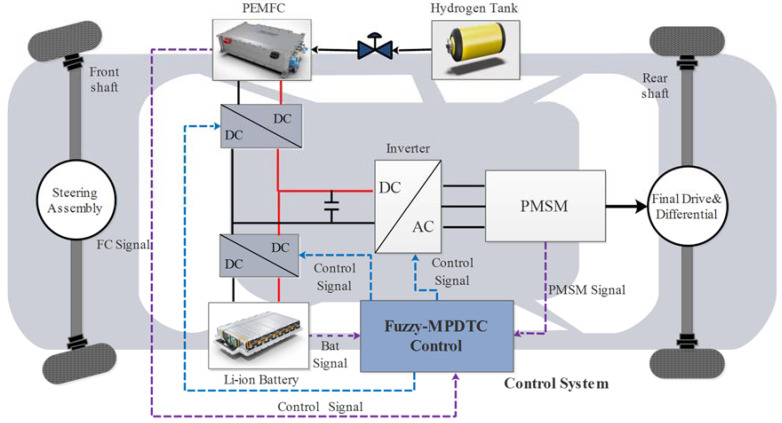
Electric vehicle configuration.

**Figure 2 sensors-22-05669-f002:**
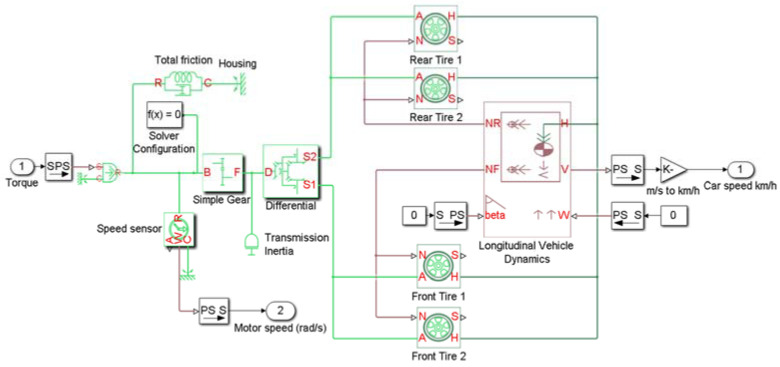
Vehicle dynamics system.

**Figure 3 sensors-22-05669-f003:**
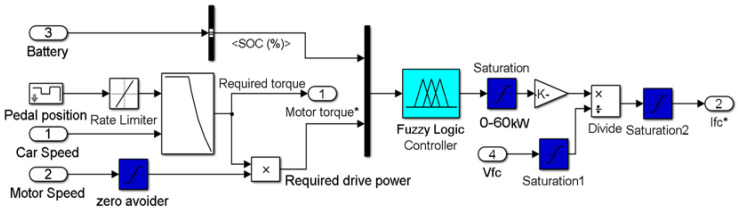
The block diagram of the fuzzy logic-based energy management strategy.

**Figure 4 sensors-22-05669-f004:**
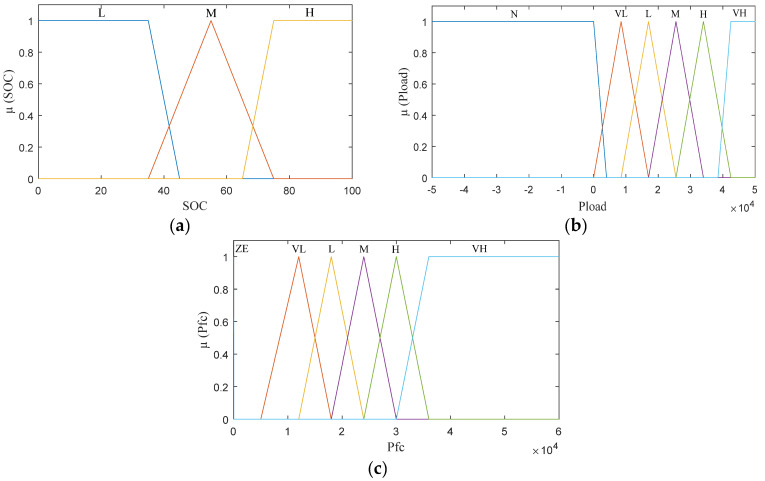
Membership functions: (**a**) input parameter (SOC); (**b**) input parameter (Pload); (**c**) output parameter (Pfc).

**Figure 5 sensors-22-05669-f005:**
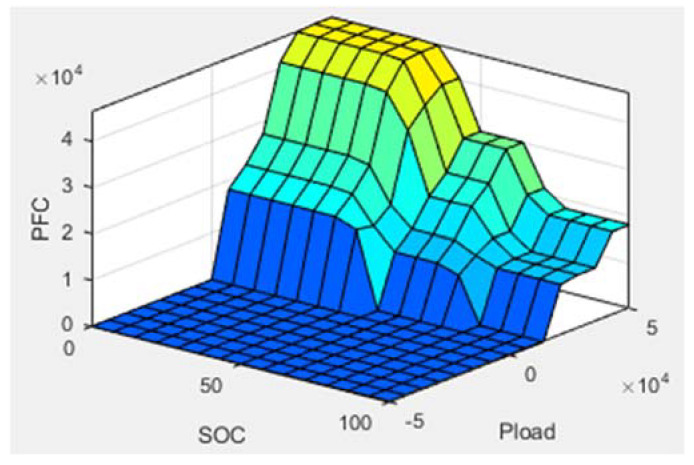
Fuzzy logic control surface.

**Figure 6 sensors-22-05669-f006:**
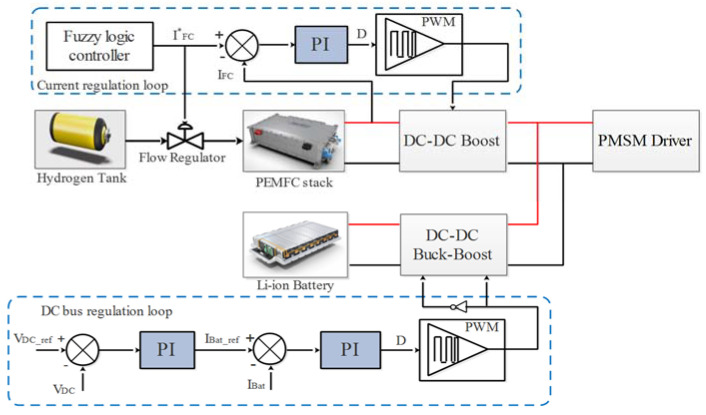
DC bus voltage regulation, Li-ion Battery converter control, and PEMFC stack converter control.

**Figure 7 sensors-22-05669-f007:**
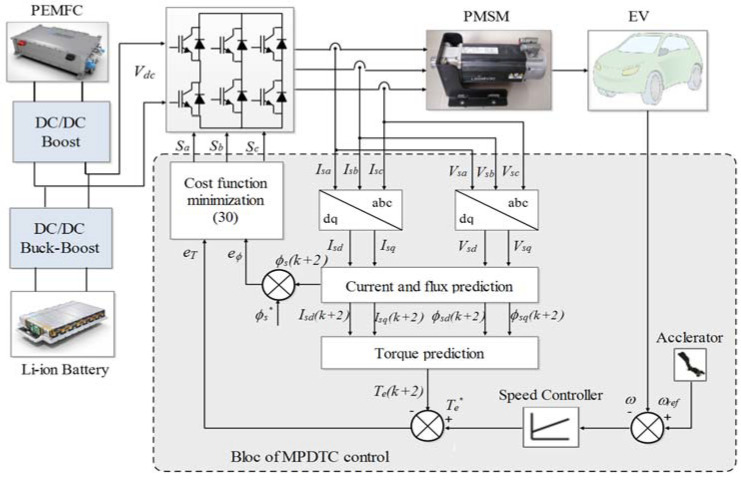
The proposed MPDTC scheme.

**Figure 8 sensors-22-05669-f008:**
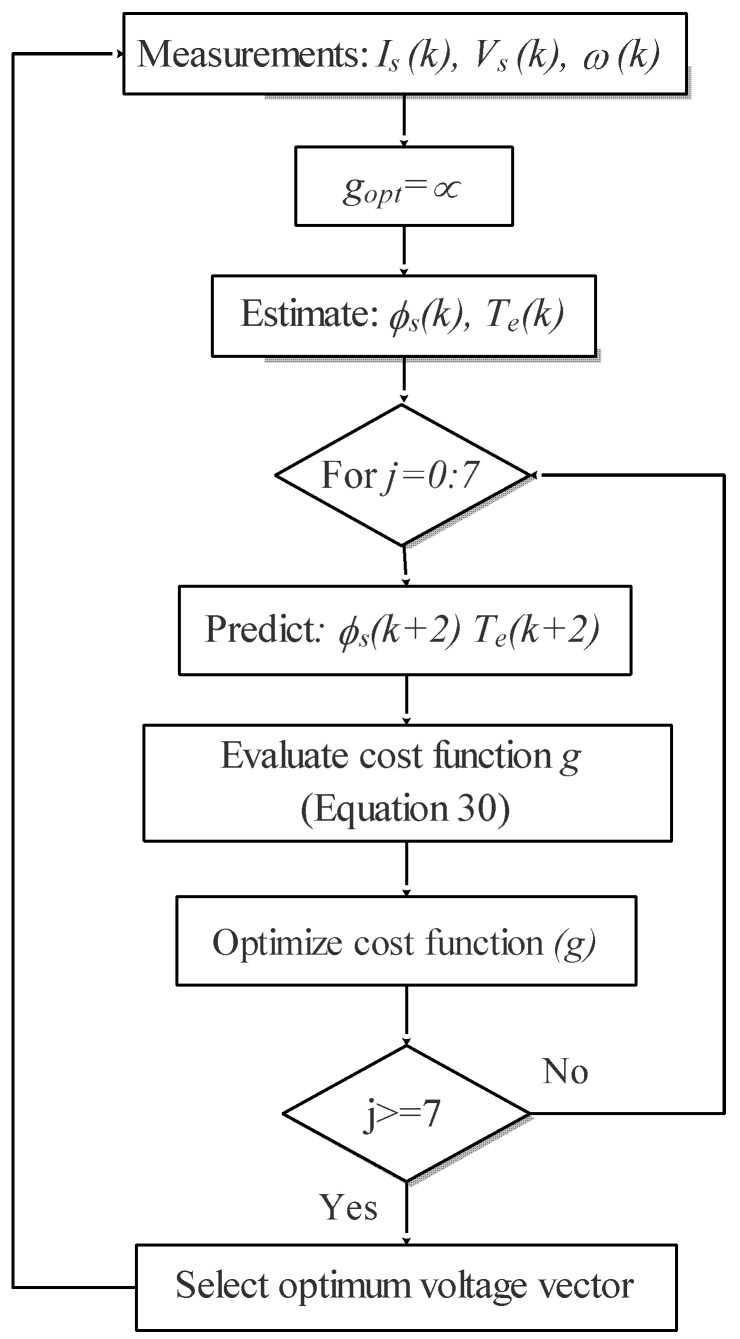
Flowchart of the proposed MPDTC.

**Figure 9 sensors-22-05669-f009:**
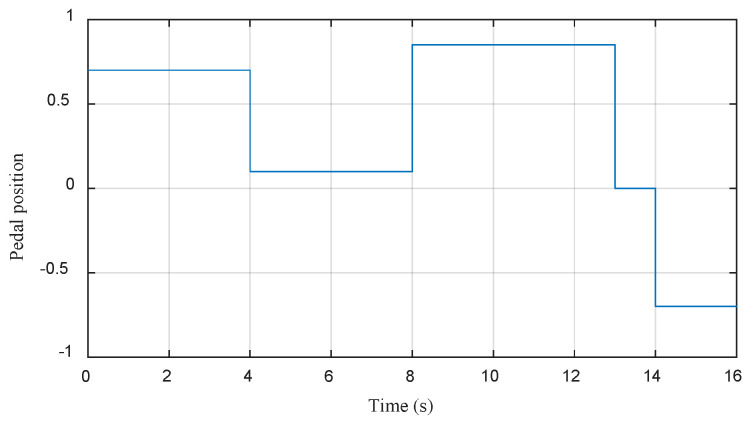
The acceleration pedal positions.

**Figure 10 sensors-22-05669-f010:**
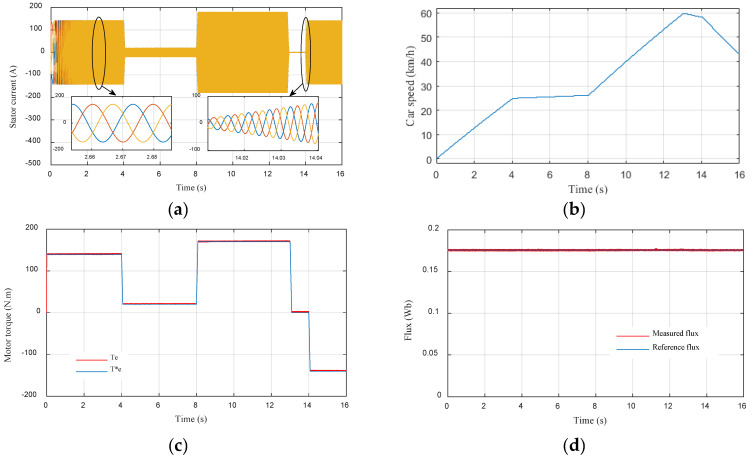
Performances of the EV traction chain under different driving modes using the MPDTC strategy (**a**) Stator current of the PMSM; (**b**) Vehicle speed response; (**c**) Electromagnetic torque of the PMSM; (**d**) Stator flux.

**Figure 11 sensors-22-05669-f011:**
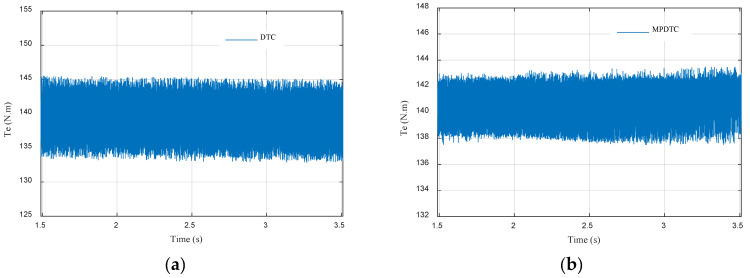
Electromagnetic torque ripples of (**a**) DTC; (**b**) MPDTC.

**Figure 12 sensors-22-05669-f012:**
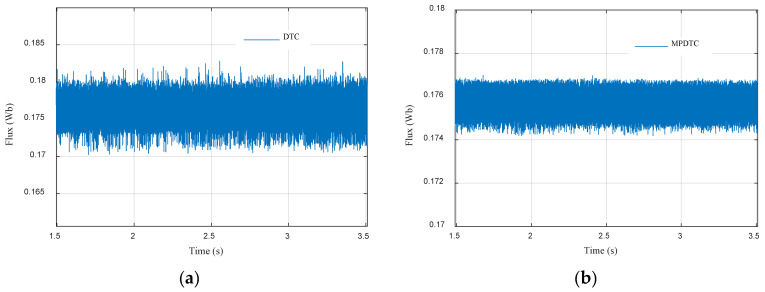
Stator flux ripples of (**a**) DTC; (**b**) MPDTC.

**Figure 13 sensors-22-05669-f013:**
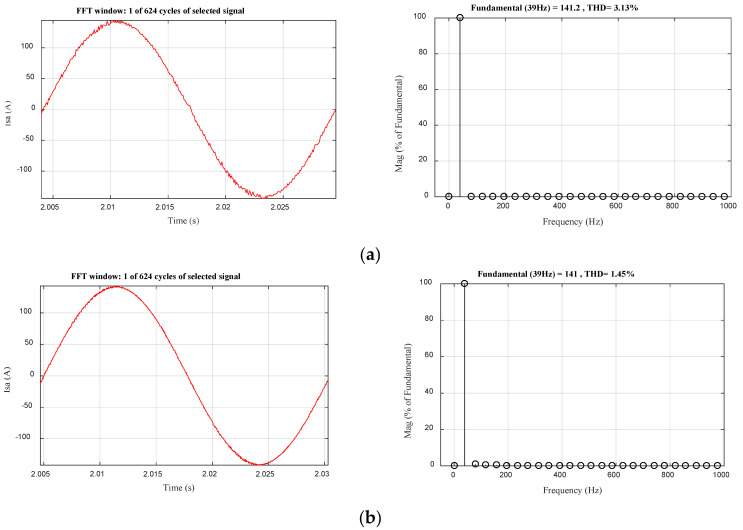
FFT analyses of phase current (**a**) DTC; (**b**) MPDTC.

**Figure 14 sensors-22-05669-f014:**
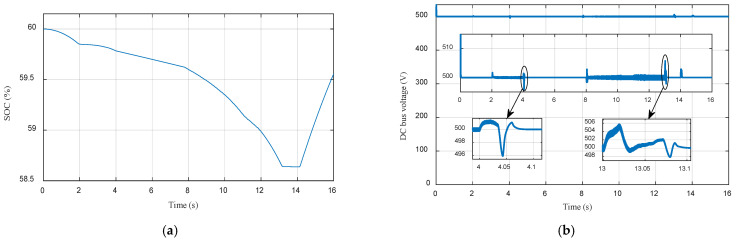
(**a**) Li-ion battery state of charge; (**b**) DC bus voltage.

**Figure 15 sensors-22-05669-f015:**
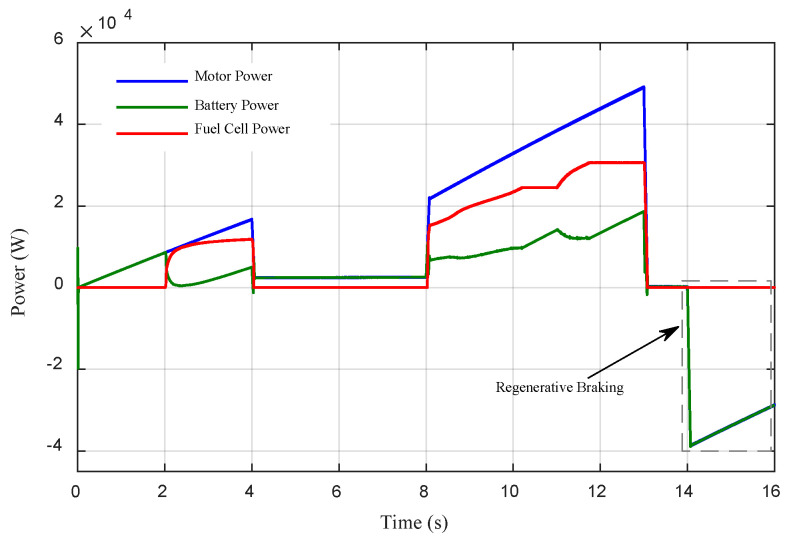
The power management of the electric vehicle under different driving modes.

**Figure 16 sensors-22-05669-f016:**
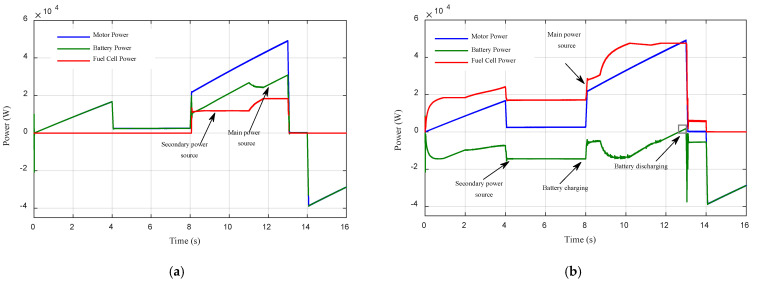
The performance of the electric vehicle under different states of charge: (**a**) SOC = 80%; (**b**) SOC = 30%.

**Figure 17 sensors-22-05669-f017:**
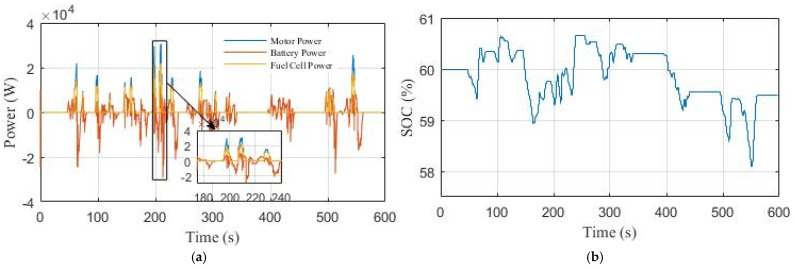
The performance of the EV under NYCC driving cycle in Scenario 1: (**a**) power curves; (**b**) Li-ion battery SOC curve.

**Figure 18 sensors-22-05669-f018:**
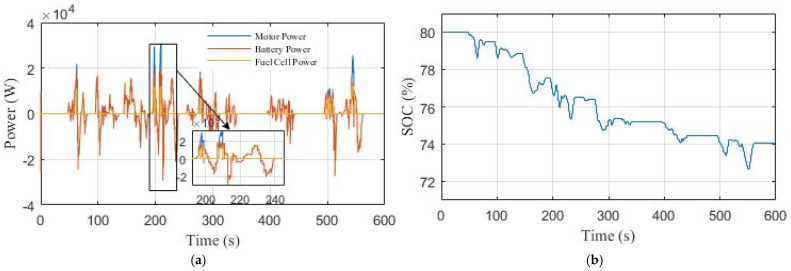
The performance of the EV under NYCC driving cycle in Scenario 2: (**a**) power curves; (**b**) Li-ion battery SOC curve.

**Figure 19 sensors-22-05669-f019:**
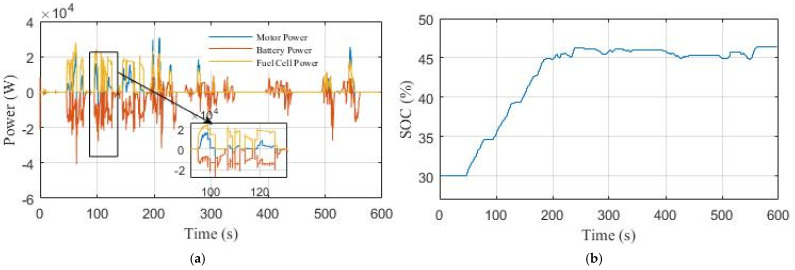
The performance of the EV under NYCC driving cycle in Scenario 3: (**a**) power curves; (**b**) Li-ion battery SOC curve.

**Figure 20 sensors-22-05669-f020:**
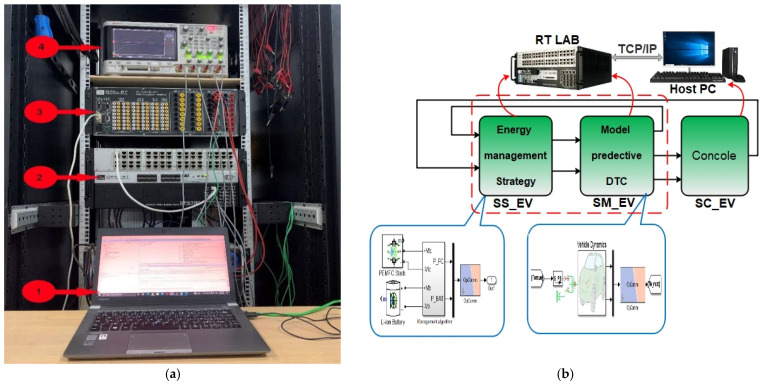
(**a**) Experimental setup of RT-lab platform at LTII laboratory; (**b**) RT-lab system architecture.

**Figure 21 sensors-22-05669-f021:**
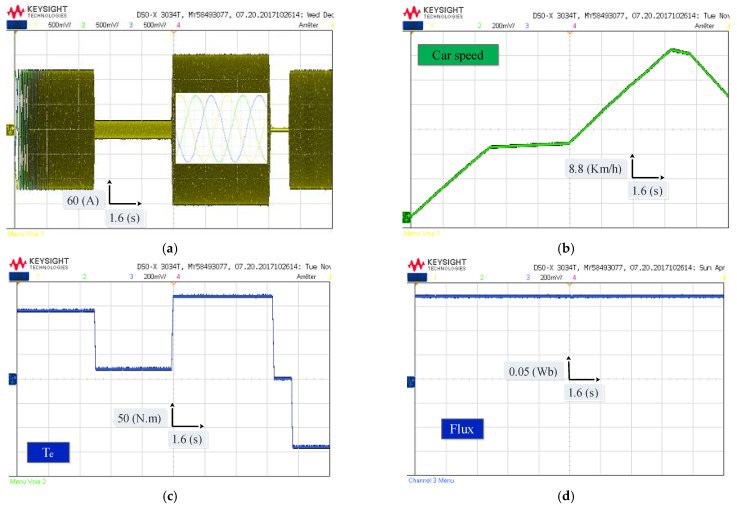
Experimental results of the EV traction chain under different driving modes using the MPDTC strategy (**a**) Stator current; (**b**) Vehicle speed response; (**c**) Electromagnetic torque; (**d**) Stator flux.

**Figure 22 sensors-22-05669-f022:**
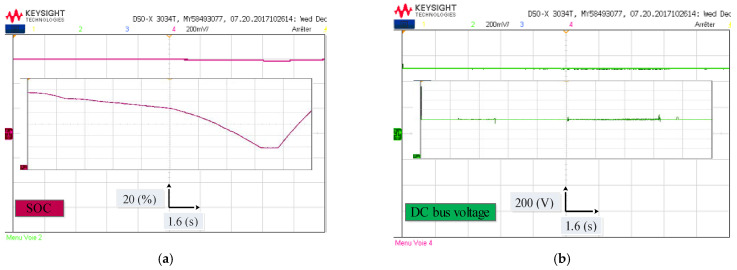
Experimental waveform under different driving modes (**a**) Battery state of charge; (**b**) DC bus voltage.

**Figure 23 sensors-22-05669-f023:**
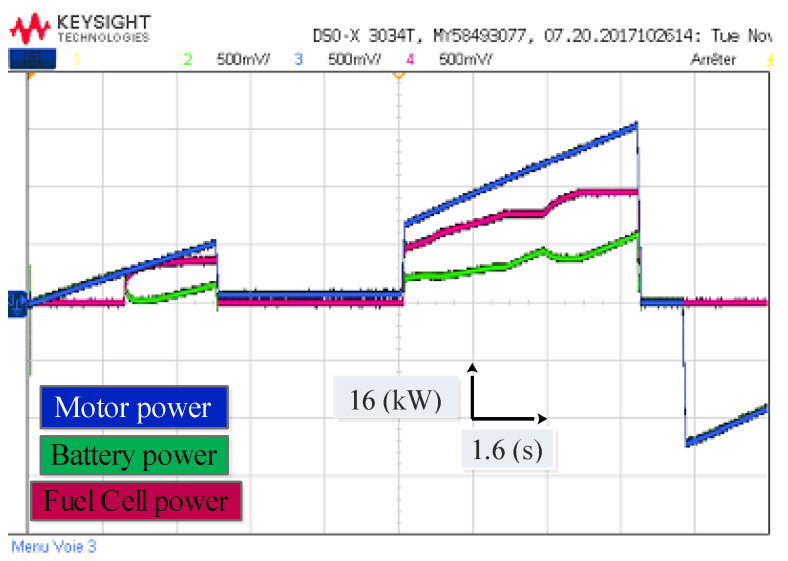
Experimental waveform of the power management of the electric vehicle under different driving modes.

**Figure 24 sensors-22-05669-f024:**
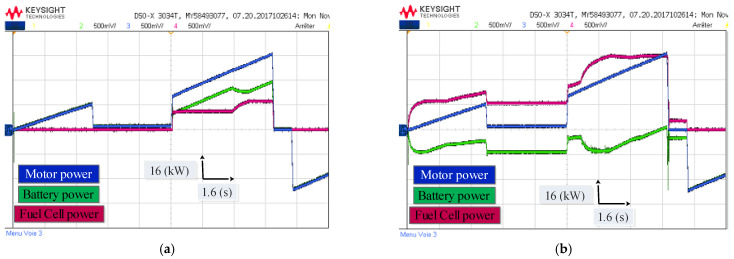
Experimental waveform of the performance of the electric vehicle under different state of charge (**a**) SOC = 80%; (**b**) SOC = 30%.

**Table 1 sensors-22-05669-t001:** PEMFC stack parameters.

Parameter	Value	Unite
Nominal power	50	kW
Peak power	60	kW
Number of cells	358	Cell
Nominal stack efficiency	55	%
Operating temperature	65	°C
Nominal Air flow rate	2100	Ipm
Fuel supply pressure	1.5	bar
Air supply pressure	1	bar

**Table 2 sensors-22-05669-t002:** PMSM parameters.

Parameter	Value	Unite
$\mathrm{Rated}\;\mathrm{power}\;(P_{r}$)	50	kW
$\mathrm{DC}\;\mathrm{voltage}\;(V_{dc}$)	500	V
$\mathrm{Stator}\;\mathrm{resistance}\;(R_{s}$)	0.0065	Ω
$\mathrm{Stator}\;\mathrm{inductance}\;(L_{d}$$,\;L_{q}$)	8.35	mH
$\mathrm{PM}\;\mathrm{magnetic}\;\mathrm{flux}\;(\phi_{f}$)	0.17566143	Wb
Number of pole pairs (*p*)	4	-
Motor inertia (*J*)	0.089	kg·m^2^
Viscous damping (*f*)	0.005	N·m·s

**Table 3 sensors-22-05669-t003:** Electric vehicle parameters.

Parameter	Value	Unite
Vehicle total mass (M)	1325	kg
Gear ratio of the final drive (G)	5.2	-
Number of wheels per axle	2	-
$\mathrm{Frontal}\;\mathrm{area}\;(A_{f}$)	2.57	m^2^
Tire radius (r)	0.3	m
$\mathrm{Drag}\;\mathrm{coefficient}\;(C_{d}$)	0.3	-

**Table 4 sensors-22-05669-t004:** Energy management fuzzy logic rules.

*P_fc_*		*P_load_*					
		N	VL	L	M	H	VH
** *SOC* **	**L**	ZE	L	M	H	VH	VH
	**M**	ZE	ZE	VL	L	M	H
	**H**	ZE	ZE	ZE	VL	VL	L

**Table 5 sensors-22-05669-t005:** Performances of the proposed MPDTC and conventional DTC in terms of torque ripples, flux ripples, and THD.

Performances	DTC	MPDTC	Improvement (%)
Torque ripples (N·m)	140 ± 11	140 ± 5	54.54
Flux ripples (Wb)	0.175 ± 0.01	0.175 ± 0.0023	77
THD (%)	3.13	1.45	53.67

## Data Availability

Not applicable.

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
