# Peer review of "Model Predictive Direct Torque Control and Fuzzy Logic Energy Management for Multi Power Source Electric Vehicles"

_sensors, 2022, doi:10.3390/s22155669_

Round 1
Reviewer 1 Report
The article is made at a sufficiently high scientific level, is well presented, and does not contain errors. However, due to the large volume of the object taken for research, the applied methods and their processing are not deep enough. So, for example:
1. The EMS used in the work is very simple, as it performs the function of distributing energy flows between FC and B based on two input variables - battery SOC and power of the drive. At the same time, even in the literature review, there are references in which much more complex criteria are applied, for example, ensuring the minimum cost of a hybrid power system or increasing the service life of FC or B.
2. As for PMSM control, the shown in Fig. 7 MPDTC scheme proposed does not show that flux weakening is implemented for operation in the constant power region, which is important for electric vehicles
3. The implementation of the Vehicle Dynamics System is not described at all.
4. It is almost not disclosed how the models are built for real-time simulation using the RT LAB platform. Judging by the complete identity of the time diagrams obtained during simulation in the Matlab/Simulink environment and on the RT LAB platform, the same models of all elements of the electric vehicle were used. This raises doubts about the expediency of using real-time simulation. For verification of mathematical modeling, it would be much better to apply a HIL study of at least one of the considered subsystems.
Other minor notes:
In line 155, there is a citation [38] with motor parameters, but the indicated highly cited work, which is devoted to the modeling of different types of batteries, does not contain any data on the parameters of the motor of an electric vehicle.
As indicated in Table. 2 parameter of PMSM Фf is not correctly named. There should be not just a flux linkage, but also a flux linkage from the PM or, more precisely, from a pair of PM poles.
Author Response
Dear Reviewer,
Authors are thankful to the honorable Editor-in-chief and the honorable Editor for considering our paper and for giving us an opportunity to incorporate the pertinent suggestions given by the honorable reviewers. We hope the answers and changes were sufficient.
A more detailed list of changes and responses to review comments can be found in the attached document.
Sincerely
Vojtech Blazek

Reviewer 2 Report
My comments are:
1. Please give some quantified results/numbers in the abstract. "Effectiveness" is too vague. Same for conclusions.
2. Lines 90-100: be consistent with the tense. The authors switch from present to past and it is confusing. The whole paper must be in the past tense.
3. Line 130: delta G, please fix.
4. What is the configuration of the fuel cell stack (m x n x p?).
5. Symbols should be one letter with a subscript if needed. 'SOC' is not a good symbol. It could be mistaken as three symbols, S, O, and C.
6. Block diagram, not bloc.
7. Fuzzy acronyms must be expanded.
8. Line 213: shown.
9. Authors repeatedly include the figure before the explanation. Change it.
10. I cannot recommend the publication of these results when the simulation of a vehicle's performance lasted only 16 s. This is not a realistic time frame. Must longer time frames must be shown in order to explore the effect of using up the entire fuel supply of the vehicle.
11. Very little comparison to existing studies is done. It's not sufficient to propose a new system, which itself is heavily derivative of existing systems. For all this complexity and cost, what percent improvement is gained over cited studies? Explain in your results what fatal flaw has been mitigated. Give the reader a reason to adopt your methods. There are literally hundreds of EMS studies in the past 10 years that are indistinguishable from this one. Several of them have even been cited in this work.
Author Response

(The authors gave the same response as above.)

Round 2
Reviewer 2 Report
The authors have taken my numerous remarks very seriously and have striven to improve their paper. I am aware that 'SOC' is used as a symbol for state of charge, but this is a widespread error nonetheless. Giving examples of previous errors is no defense. I won't press the matter, because it is minor. But I ask the very astute authors to consider this. Of all the dozens to hundreds of symbols and quantities in their paper, every single one follows the rule of one character + subscript or superscript - except for 'SOC'. Why is this one special?
Anyway, I can recommend this current version for publication. Thanks.